# Metabolic Syndrome-Related Knowledge, Attitudes, and Behavior among Indigenous Communities in Taiwan: A Cross-Sectional Study

**DOI:** 10.3390/ijerph20032547

**Published:** 2023-01-31

**Authors:** Shu-Fen Lo, Fang-Tsuang Lu, An-Chi O. Yang, Jia-Ling Zeng, Ya-Yu Yang, Yen-Ting Lo, Yu-Hsuan Chang, Ting-Hsuan Pai

**Affiliations:** 1Department of Nursing, Tzu Chi University, Hualien 970374, Taiwan; 2Charity Development Department, Tzu Chi Charity Foundation, Hualien 971067, Taiwan; 3School and Graduate Institute of Nursing, National Taiwan University, Taipei 100233, Taiwan; 4Dianthus Medical Group, Taoyuan 320032, Taiwan; 5Department of Nursing, Taipei Guang En Elderly Medicare Center, New Taipei City 231040, Taiwan; 6Nursing Department, Lo-Hsu Medical Foundation, Lotung Poh-Ai Hospital, Yilan 265501, Taiwan; 7Nursing Department, Far Eastern Memorial Hospital, New Taipei City 220216, Taiwan

**Keywords:** metabolic syndrome, knowledge, attitude, self-management behavior, indigenous

## Abstract

Background: Metabolic syndrome is characterized by cardiovascular and chronic disease risk factors that cause health problems. Inequalities in medical resources and information present a challenge in this context. Indigenous communities may be unaware of their risk for metabolic syndrome. Aims: This study explored factors associated with metabolic syndrome-related knowledge, attitudes, and behaviors among Taiwanese indigenous communities. Methods: For this descriptive cross-sectional survey, we collected anthropometric data and used a self-administered questionnaire between 1 July 2016, to 31 July 2017, from a convenience sample of an indigenous tribe in eastern Taiwan. The response rate was 92%. Results: The prevalence of metabolic syndrome was as high as 71%, and the average correct knowledge rate was 39.1%. The participants’ self-management attitudes were mainly negative, and the self-management behaviors were low in this population. Stepwise regression analysis showed that knowledge, attitude, age, perception of physical condition, and body mass index, which accounted for 65% of the total variance, were the most predictive variables for self-management behaviors. Conclusions: This is the first study to report the relationship between metabolic syndrome knowledge, attitudes, and behaviors in an indigenous population. There is an urgent need to develop safety-based MetS health education programs that can provide access to the right information and enhance self-management approaches to lessen the growing burden of MetS in indigenous communities.

## 1. Introduction

The top 10 causes of death in Taiwan and in the Asia-Pacific region have changed from infectious to noncommunicable disease, particularly cardiovascular diseases (CVDs) and diabetes mellitus (DM) [1,2]. The causes of death are mainly chronic diseases, such as CVDs, DM, nephropathy, hypertensive diseases, and cancer [1]. These chronic diseases accounted for 31.4% of total deaths in Taiwan in 2020 [2]. Metabolic syndrome (MetS), characterized by CVDs and chronic disease risk factors, can cause health problems [1]. Compared with the healthy population, patients with CVDs and DM are 2 and 3.5–5 times more likely to develop MetS [3]. MetS is currently a global burden on healthcare systems and a serious public health issue [4,5]. According to modified ATP ΙΙΙ criteria, MetS is defined as the presence of at least three of the following component risk factors indicating the MetS occurrence: (1) waist circumference (WC) >90 and >80 cm for men and women; (2) systolic blood pressure (SBP) and diastolic blood pressure (DBP) ≥130 mmHg and ≥85 mmHg, or current use of antihypertensive drugs; (3) fasting plasma glucose (FPG) level ≥110 mg/dL or current use of antihyperglycemic drugs; (4) either triglyceride (TG) level ≥150 mg/dL or current use of TG-lowering medication; and (5) high-density lipoprotein (HDL) level <40 mg/dL and <50 mg/dL for men and women or current use of lipoprotein-lowering medication [1,3]. Increasing economic development and aging in Asia are major contributors to the increasing prevalence of MetS [1]. MetS’s prevalence is 24.5% in China [6] and 31.7% in Korea [7]. In Taiwan, its prevalence ranges from 6.8% to 42.9% [8,9]. MetS can occur due to a sedentary lifestyle, including insufficient exercise, imbalanced diet (e.g., with high fat), and obesity [10].

MetS’s prevalence is currently higher in indigenous communities than in nonindigenous populations; for example, its prevalence in indigenous populations is 53.1% in Baja California, Mexico [4], 37.5%–66.1% in Brazil [11], and 42.9% in Taiwan [10]. Most indigenous communities live in remote mountainous areas that lack medical-care resources. Their lifestyles also differ from those of the other populations. Over the past 30 years, the prevalence of hypertension and DM and the standardized death ratio have increased among the Taiwanese indigenous communities [9,12]. Thus, MetS-related factors and the health effects on these indigenous communities should be emphasized. In 2019, the average life expectancy of an indigenous person was 73.10 years (7.76 years lower than a nonindigenous person), and that of a mountain-dwelling indigenous person was even lower (by 9.52 years) [13].

The information–motivation–behavioral skills model measures one’s knowledge, attitude, and self-management behavior (KAB), the key factors affecting one’s propensity to develop MetS [14,15]. Indigenous communities have limited access to healthcare information because of their unique cultural background, living environment, inadequate medical and sanitation conditions, and remoteness [9]. In Taiwan, older age, female, unhealthy dietary patterns, insufficient exercise, lower educational level, alcohol consumption, and betel quid chewing are potential risk factors for MetS [12,16]. Although the effects of inadequate behavior regarding DM and CVDs and their prevalence have been emphasized [10], studies on MetS in indigenous individuals have only focused on the causes of death, particularly disease prevalence, and risk factors for MetS [4,9,17,18]. In light of this background, this study had two main aims: (a) to assess the level of knowledge, attitudes, and behavior of Taiwanese indigenous people towards MetS, and (b) to determine factors predicting MetS self-management behavior among this population.

## 2. Materials and Methods

### 2.1. Study Design and Setting

This cross-sectional study recruited three major groups (Ami, Bunun, and Taroko) from eight indigenous tribe areas in eastern Taiwan. Data were collected from 1 July 2016, to 31 July 2017. The inclusion criteria were as follows: (1) age > 20 years, (2) indigenous people living in tribes, (3) ability to read the questionnaire, and (4) willingness to participate. The exclusion criterion was the presence of a mental illness. We used the convenience sampling method. Potential subject recruitment was carried out through tribal council chairpersons or churches. Participants were informed verbally and in writing of the study’s purpose and their rights, including anonymity, and that participation was voluntary. The time needed for MetS anthropometric measurement and to complete the questionnaire was about 15–20 min.

### 2.2. Sample Size

To determine an adequate sample size and avoid type II errors, G*Power 3.1.7 free- software was used. Estimates were performed using G-Power 3.1, with a set power of 0.80, a medium effect size of 0.3, and α of 0.05, and analyzed using stepwise regression. Thus, the number of suggested required samples was 208. A 20% churn rate was used to calculate the number of samples, which was 250 subjects.

### 2.3. Ethical Considerations

All included participants provided informed consent. The Ethics Committee and Community Review Board (CRB No. CRB-105-12) of the Hualien Tzu Chi Hospital (IRB No. IRB105-58-A) approved this study.

### 2.4. Instruments

A demographic data sheet was used to record the participant data. The five instruments used are described below.

#### 2.4.1. Demographic Data

Sociodemographic data were collected, including age, sex, marital status, educational level, occupation, perceived physical condition (PPC), and medical history of hypertension, diabetes, CVD, and kidney disease.

#### 2.4.2. Anthropometric Measurement

We used the definition for MetS of the Health Promotion Administration, Ministry of Health, and Welfare, Taiwan, based on the modified ATP III and MetS criteria [1,3]. The screening process took 10 min, and our research team conducted measurements as follows: each participant’s height and weight were determined to the nearest 0.1 cm and 0.1 kg, respectively, using a mobile combined scale and stadiometer (Health team, Georgia, USA), respectively. WC was measured to the nearest 0.1 cm using a standard, no-stretch measuring tape (Gulik, Lafayette Instrument Co., Lafayette, IN, USA). Omron HBF-251 was used to measure body mass index (BMI). Blood pressure (BP) was measured using a 360° full-covering cuff digital BP monitor (Omron 7 series Plus) in a seated position after a 10-min rest, with participants’ arms relaxed and palms facing upward. Finally, CardioChek-P-A was used to measure FPG, HDL, and TG levels.

#### 2.4.3. MetS Knowledge Scale (MetS-KS)

The MetS Knowledge Scale (MetS-KS), developed by See et al. [19] and Hung et al. [12], comprises 14 items in a single selection format (to prevent participants from just guessing the answer, thus improving the response credibility); each response is assessed on a five-point scale. A score of 1 indicates the correct answer, and 0 the incorrect answer. The higher the score, the higher the MetS knowledge. After the factor analysis, MetS defection, MetS prevention, and MetS-related CVD domains were extracted, which explained 69.61% of the variance, and Cronbach’s α was 0.82.

#### 2.4.4. MetS Attitude Scale (MetS-AS)

The MetS-AS, developed by See et al. and Hung et al. [12,19], was used, which comprised 13 items; each item is scored on a five-point Likert scale (from 1, strongly disagree, to 5, strongly agree; total score range, 13–65). The higher the score, the more positive the MetS attitude. After factor analysis, MetS self-control and lifestyle domains were extracted, which explained 50.99% of the variance, and Cronbach’s α was 0.87.

#### 2.4.5. MetS Self-Management Behavior Scale (MetS–SMBS)

The MetS–SMBS, developed by See et al. and Hung et al. [12,19], was used. The scale comprises seven items reviewing an individual’s health behavior frequency in the past week. Each item is scored on a five-point Likert scale (from 1, strongly disagree, to 5, strongly agree; total score range, 7–35). The higher the scores, the higher the level of self-management behavior. After factor analysis, MetS self-control and lifestyle domains were extracted, which explained 62.58% of the variance, and Cronbach’s α was 0.77.

### 2.5. Data Analysis

Data were analyzed using SPSS for Windows (version 21.0; SPSS Inc., Chicago, IL, USA). Descriptive statistics assessed the demographic and anthropometric characteristics. One-way analysis of variance (ANOVA) was used to verify differences in MetS–SMBS scores based on the general characteristics. Post hoc analyses were performed using Scheffe’s test. Pearson correlation analysis was used to analyze the correlation among MetS-KS, MetS-AS, and MetS–SMBS scores. Finally, multiple regression analysis identified factors influencing MetS–SMBS scores. For all tests, a *p*-value < 0.05 was considered statistically significant.

## 3. Results

In total, 250 participants were recruited, and 231 responses were deemed eligible for inclusion in the analysis. Those responses excluded 10 that were incomplete and 9 that did not meet the eligibility criteria. The completion rate of the survey was 92%.

### 3.1. Participants’ Characteristics

The sociodemographic characteristics of the respondents are presented in Table 1. Our participants’ mean age was 55.75 ± 14.02 (22–79). The male-to-female ratio was 1:1.25. The highest educational level for most participants was senior high school (35.9%). Moreover, 68.7% were married. The number of chronic diseases ranged from 0 to ≥4; 63.6% of participants had more than one chronic disease (mean, 1.13 ± 1.21), and 62.3% of participants thought their health condition was common. Nearly 60% of participants were obese. Table 2 shows the anthropometric and clinical characteristics of the population under study. The abnormality rate was 66.5% for those exceeding 3 MetS indicators; 31.0% of participants were at high risk for MetS. The proportion of males with abnormal diastolic blood pressure, triglycerides, and high-density fat was higher than that of females.

### 3.2. Descriptive Analysis

Table 3 presents the scores of all scales for MetS KAB. The MetS-KS score range was 0–10 (mean, 3.91 ± 2.99), and the correct answer rate was only 39.1%. Moreover, the MetS-AS score range was 13–65 (mean, 38.97 ± 10.69), indicating a negative attitude toward MetS; however, there were large differences among participants. Finally, the overall mean MetS–SMBS scores were 18.58 ± 6.85, indicating the need for improvement in self-management behavior to control MetS incidence. Females’ attitude (t = 2.04, *p* = 0.039) and behavior scores (t = 2.37, *p* = 0.015) were higher than males’, showing statistical significance.

### 3.3. Relationships between Independent Variables and Self-Management Behavior

One-way ANOVA analyzed factors influencing MetS–SMBS scores. After a post hoc Scheffe test was performed, the MetS–SMBS scores of female participants were higher than those of male participants (F = 6.003, *p* = 0.015). Relationships among demographic variables, MetS-KS scores, MetS-AS scores, and MetS–SMBS scores were analyzed and significantly correlated with MetS knowledge, MetS attitude, age, TG, number of chronic diseases, and DBP (*p* < 0.05; Table 4). These results indicate that the better the participant’s MetS knowledge, the more positive their MetS attitude, and that those participants with normal DBP and TG levels and few chronic diseases are less likely to develop MetS.

### 3.4. MetS Self-Management Behavior Predictors

As presented in Table 5, multiple regression analysis was performed on all independent variables; analyses were based on the significance of our Pearson correlation analysis results. Discontinuous variables were converted into dummy variables before the analysis. The results reveal that MetS-AS scores, MetS-KS scores, age, PPC, and BMI were all significant variables determining MetS–SMBS scores, accounting for 65% of the total variance. MetS-KS and MetS-AS scores strongly predicted MetS–SMBS scores in participants.

## 4. Discussion

### 4.1. Main Findings

This was the first study to report the relationship between MetS knowledge, attitude, and self-management behavior in indigenous communities. The MetS prevalence was 66.5%, and most participants had insufficient MetS knowledge, too few MetS self-management behaviors, and negative MetS attitudes. Moreover, the association between behavior and biochemical and anthropometric variables was revealed. Increasing age was associated with higher knowledge scores and a more positive attitude toward reducing MetS risk, and good PPC and an ideal BMI were shown to improve the MetS self-management behavior in these indigenous individuals. The MetS prevalence was higher in women than in men.

### 4.2. Participant Characteristics

This study found that in a rural indigenous community in eastern Taiwan, 66.5% of the population had MetS, and 31.0% had a high MetS risk, with a higher prevalence in women. This prevalence is higher than that reported in Taiwan, whether for indigenous or nonindigenous populations [9,20,21,22]. The MetS prevalence rate was 58.7% in an indigenous community in southeastern Taiwan [20], 48.3% in 174 indigenous people in northern Taiwan [23], and 35.0% in 2596 middle-aged and elderly people in southern Taiwan [21]. Moreover, the MetS prevalence among the eastern Taiwanese indigenous communities noted in this study was much higher than that of indigenous communities in Brazil (66.1%) [11], Mexico (53.1%) [4], Malaysia (29.57%) [18], and Australia (50.3%) [24].

The population in our study also lives in remote and mountainous areas, with limited access to medical-care resources. The indigenous people also seldom use healthcare resources in public health centers or mountain-tour medical services, or visit the hospital, unless they are sick. High-MetS-risk groups and people with MetS are thus difficult to track through local health screening service and hospital-based sampling. Thus, active care can be used as a health educational intervention at night or during leisure time. These can reflect the average indigenous person’s perception of MetS and help enhance the native community’s understanding of MetS.

### 4.3. MetS Knowledge, Attitude, and Self-Management Behavior

These results indicate that the correct answer rate was 39.1% on the MetS-KS administered to eastern Taiwanese indigenous individuals. More than half of the participants had MetS but lacked an adequate understanding of the definition and diagnosis of MetS and could not understand whether they had MetS or high DM or CVD risk. These results were much lower than that reported by See et al. [19] (44.2% in 88 participants). Similarly, Amarasekara et al. [25] surveyed 423 participants in Sri Lanka and found them to have moderate MetS knowledge. In general, a participant’s MetS knowledge is significantly associated with MetS occurrence; thus, the better a community’s knowledge of MetS, the more easily it is able to control the MetS occurrence [12]. Our mean MetS-KS scores were low because 53.7% of our participants, who belonged to a rural tribe, had an educational level of sixth grade or lower (i.e., junior high school or lower). They had fewer opportunities than the general population to obtain MetS information. Whether from a health center or mountain medical tour, medical personnel conduct health education daily. However, if medical personnel arrive during the working hours of indigenous people, they may not easily obtain health information. Thus, we suggest that relevant authorities should consider the geographical environment and work habits of indigenous cultures when designing MetS programs for these communities. For effective health education, medical personnel should conduct health education and health screening activities during tribal celebrations, such as various ball games, harvest festivals, and piercing festivals in villages and townships.

Patients with a positive attitude and better health behaviors have better disease control [1]. The health attitude of participants may affect their self-management behaviors for chronic diseases [9,14]. Alefishat et al. [14] investigated 990 participants who had a high MetS risk in Jordan and found that participants had a negative attitude toward MetS. This significantly affected their self-management behaviors in response to MetS. The participants’ MetS attitude also tended to be negative in this study, potentially due to poor economic conditions, making it difficult to regularly monitor residents’ BP, blood sugar, and body weight. We recommend that administrative agencies set up equipment, such as sphygmomanometers and weighing scales, in the churches and grocery stores of indigenous communities. Through proper health education, medical personnel can also improve indigenous communities’ lifestyle and reduce their MetS risk [26].

Healthy self-management behaviors, such as adequate physical activity, proper diet, and reduced alcohol consumption, can reduce MetS incidence [27]. Hung et al. [12] investigated 725 indigenous participants in Chiayi, Taiwan. They found that age, educational level, drinking behavior, and betel nut chewing had significant effects on MetS risk; the authors reported that 72.3% of participants consumed alcohol every 1 or 2 days, but up to 50% of participants consumed alcohol daily. These participants also had insufficient exercise frequency, with most participants considering work exercise. Factors associated with MetS lifestyle depend on the environment, getting health messages, and sufficient motivation to engage in healthy behaviors and skills [15,27]. We found that the majority of the tribe’s residents were farmers. These residents believed that due to the strenuous nature of their job, they must consume more salt and oil for energy. In total, 62.1% of participants had a negative attitude toward reducing their salt intake. Therefore, when performing health education activities, medical personnel should consider the indigenous community’s cultural background and work style [26]. Sports should also be taught, and a safe, friendly sports environment should be nurtured in the community. Moreover, community members were asked to walk on their community grounds together after meals. Furthermore, calorie counts should be marked for common meals provided to the community to promote healthy eating.

### 4.4. Factors Influencing MetS Self-Management Behavior

One study indicated that indigenous individuals prefer flavorful, high-salt, and high-oil diets. Thus, shifting their lifestyle in terms of dietary knowledge and behavior to one that reduces their MetS risk can be difficult [9,19]. Dietary and exercise interventions can effectively reduce BP, WC, cholesterol, and TG levels [3]. Therefore, an attitude toward promoting regular exercise, a healthy diet, and adoption of healthy living habits should be instilled in students from a young age. BP should be measured regularly for high-risk groups, and self-management behaviors should be promoted. Our study found that the participants most likely to have adequate MetS knowledge and a better attitude toward MetS prevention habits were elderly individuals who were aware of their health condition, had lower TG levels and DBP, and had healthier MetS self-management behaviors [5]. Thus, MetS knowledge and attitude are the main factors influencing MetS behavior. Indigenous cultures have differences in diet, lifestyle, working style, drinking behavior, and religious beliefs [3]. Sports and dietary programs can alleviate MetS in elderly and middle-aged women [28]. Therefore, we suggest educating indigenous communities on the benefits of exercise based on the age of individuals. We also suggest that cooperation with the local church should be the basis for establishing a culturally appropriate sports prescription training course [3].

### 4.5. Limitations

The cross-sectional design of this study did not allow the inference of causation. Furthermore, our demographic data did not include the work-type exercise behavior and dietary patterns; future studies should investigate how these factors influence the MetS risk. Moreover, our survey only investigated indigenous communities in eastern Taiwan. Although they account for a sizable 30.4% of Taiwan’s indigenous population, our results remain inapplicable to residents of other regions. Furthermore, our survey collected self-reported data; thus, information bias might have been introduced. Finally, our study found significant differences in WC, cholesterol, and BMI between genders; thus, the effects of gender on MetS risk should be investigated in the future. This study was performed between 2016 and 2017, and the inference of the research results is limited by time. At present, there are few relevant papers discussing MetS. The results of this study can be used as a reference for future related research.

## 5. Conclusions

This study collected the physiological information of 231 indigenous people in eastern Taiwan and administered a structural MetS questionnaire. Our results demonstrate that they have inadequate MetS knowledge, a negative MetS attitude, and fair MetS behavior. MetS knowledge and attitude were the factors most significantly affecting MetS self-management behavior. This finding is similar to what was found in the above-mentioned Chinese study [6]. Our results show that MetS was highly prevalent, especially in the female and elderly populations. Thus, educational programs for indigenous communities should be considered part of the current health management policy for adults at high risk of MetS. In particular, designing health education activities adapted to an indigenous community’s culture, living habits, educational levels, and treatment-seeking behaviors is necessary [26]. These programs potentially improve the community’s MetS knowledge, health attitude, and lifestyle, consequently reducing their CVS- and MetS-related chronic disease risks.

## Figures and Tables

**Table 1 ijerph-20-02547-t001:** Participant sociodemographic characteristics (n = 231).

Variable	Total (n = 231)		Male (n = 102)		Female (n = 129)		*p*-Value
	n	%	n	%	n	%	
Age (Mean ± SD)	55.75 ± 14.02		57.60 ± 14.09		53.29 ± 13.62		0.024
20–30	10	4.3	6	5.9	4	3.1	
31–40	26	11.3	10	9.8	16	12.4	
41–50	42	18.2	23	22.5	19	14.7	
51–60	51	22.0	24	23.5	27	20.9	
61–70	54	23.4	23	22.5	31	24.0	
>71	48	20.8	16	15.7	32	24.8	
Education							0.451
Sixth grade or below	119	51.5	46	45.1	73	56.6	
Junior high school	35	15.2	14	13.7	21	16.3	
Senior high school	53	22.9	26	25.5	27	20.9	
College and above	24	10.4	16	15.7	8	6.2	
Marital status							0.134
Unmarried	26	11.3	20	19.6	6	4.7	
Married	160	69.3	69	67.6	91	70.5	
Divorced	33	14.3	5	4.9	28	21.7	
Widowed	12	5.2	8	7.8	4	3.1	
Aboriginal group							0.613
Taroko	52	22.5	28	27.5	24	18.6	
Bunun	102	44.2	44	43.1	58	45.0	
Ami	77	33.3	30	29.4	47	36.4	
Employment status							0.136
Employed	129	55.8	68	66.7	61	47.3	
Unemployed	60	26.0	16	15.7	44	34.1	
Retired	42	18.2	18	17.6	24	18.6	
Health insurance							
Yes	11	4.8	8	7.8	3	2.3	
No	220	95.2	94	92.2	126	97.7	
Hypertension							0.157
Yes	88	38.3	32	31.4	56	43.4	
No	143	61.7	70	68.6	73	56.6	
Diabetes							0.632
Yes	46	19.9	18	17.6	28	21.7	
No	185	80.1	84	82.4	101	78.3	
Gout							0.001
Yes	23	10	17	16.7	6	4.7	
No	208	90	85	83.3	123	95.3	
Heart disease							0.006
Yes	40	17.32	10	9.8	30	23.3	
No	191	82.68	92	90.2	99	76.7	
Kidney disease							0.394
Yes	3	1.3	2	2.0	1	0.8	
No	228	98.7	100	98.0	128	99.2	
Drinking alcohol							0.813
Yes	112	48.5	60	58.8	52	40.3	
No	119	51.5	42	41.2	77	59.7	
Betel nut							0.651
Yes	88	38.1	46	45.1	42	32.6	
No	143	61.9	56	54.9	87	67.4	
Smoking							0.565
Yes	87	37.7	40	39.2	47	36.4	
No	144	62.3	62	60.8	82	63.6	
Perception of physical condition							0.771
Very bad	12	5.5	6	5.9	6	4.7	
Bad	28	12.7	9	8.8	19	14.7	
Average	148	62.3	65	63.7	83	64.3	
Good	26	11.8	14	13.7	12	9.3	
Very good	17	7.7	8	7.8	9	7.0	
Number of chronic diseases							0.365
None	84	36.4	30	29.4	54	41.9	
1	75	32.5	34	33.3	41	31.8	
2	44	19.0	23	22.5	21	16.3	
3	17	7.4	9	8.8	8	6.2	
4 or more	11	4.8	6	5.9	5	3.9	
Number of MetS indicators							0.628
0	3	1.3	1	1.0	2	1.6	
1	25	10.8	15	14.7	10	7.8	
2	39	16.9	14	13.7	25	19.4	
3	60	26	26	25.5	34	26.4	
4	57	24.7	25	24.5	32	24.8	
5	47	20.3	21	20.6	26	20.2	
BMI (kg/m^2^)							0.049
Underweight (<18.5)	2	0.9	1	1.0	1	0.8	
Normal (18.5~24.9)	45	19.5	22	21.6	23	17.8	
Overweight (25~29.9)	52	22.5	22	21.6	30	23.3	
Obesity (≥30.0)	132	57.1	57	55.9	75	58.1	

Note. BMI: Body Mass Index.

**Table 2 ijerph-20-02547-t002:** MetS-related indicators (n = 231).

Variable	Total (n = 231)			Male (n = 102)			Female (n = 129)			*p*-Value
	n	%	M ± SD	n	%	M ± SD	n	%	M ± SD	
SBP (mmHg)			136.65 (22.50)			136.47 ± 22.15			136.76 ± 22.80	0.253
Normal	99	43.0		42	43.3		58	43.3		
Abnormal	132	57.0		55	56.7		76	56.7		
DBP (mmHg)			83.88 ± 16.41			84.81 ± 16.09			83.22 ± 16.66	0.470
Normal	131	57.0		50	51.5		81	60.4		
Abnormal	100	43.0		47	48.5		53	39.6		
FPG (mg/dl)			131.53 ± 58.74			127.86 ± 68.36			133.41 ± 53.50	0.638
Normal	116	50.2		50	51.5		61	45.5		
Abnormal	115	49.8		47	48.5		73	54.5		
TG (mg/dl)			209.65 ± 134.24			224.85 ± 139.27			198.83 ± 129.98	0.151
Normal	99	42.9		37	38.1		60	44.8		
Abnormal	132	57.1		60	61.9		74	55.2		
HDL (mg/dl)			51.56 ± 35.63			45.29 ± 21.35			52.43 ± 20.42	0.012
Normal	116	50.2		47	48.5		66	49.3		
Abnormal	115	49.8		50	51.5		68	50.7		
WC (cm)			92.85 ± 13.28			93.16 ± 14.14			92.55 ± 12.70	0.735
Normal	57	24.7		31	31.9		24	17.9		
Abnormal	174	75.3		66	68.1		110	82.1		

Note. M: mean; SD: standard deviation; n: number; %: percentage; SBP: systolic blood pressure; DBP: diastolic blood pressure; FPG: fasting plasma glucose; TG: triglycerides; HDL: high-density lipoprotein; WC: waist circumference.

**Table 3 ijerph-20-02547-t003:** Mean scores on MetS knowledge, attitude, and self-management behavior scales and their domains (n = 231).

Variable (Item Number)	Total (n = 213)			Male (n = 102)			Female (n = 129)			*p*-Value
	Mean	SD	CA (%)	Mean	SD	CA (%)	Mean	SD	CA (%)	
MetS-KS (14)	3.91	2.99	39.1%	3.51	2.94	38.7%	4.07	3.12	40.3%	0.134
MetS prevention (5)	2.45	1.86	49.0 %	1.96	1.62	40.2%	2.29	1.74	49.8%	0.157
MetS cardiovascular disease (4)	1.40	1.30	35.0%	1.23	1.08	30.8%	1.68	1.28	37.0%	0.139
MetS definition (5)	0.05	0.22	5.0%	0.05	0.72	5.6%	0.24	0.65	8.2%	0.898
MteS-AS (13)	38.97	10.69		37.15	11.41		40.24	10.51		0.039
MetS control (5)	13.87	4.18		12.46	4.96		13.63	4.48		0.068
MetS lifestyle (8)	25.10	7.56		24.64	7.26		26.50	6.91		0.073
MetS-SMBS	18.58	6.85		17.85	7.51		20.10	6.19		0.015
MetS control (3)	5.96	3.30		5.86	3.52		6.48	3.31		0.008
MetS lifestyle (4)	12.61	4.79		12.00	5.01		13.70	4.46		0.010

Note. SD: standard deviation; CA (%): correct answer percentage; MetS-KS: Metabolic Syndrome Knowledge Scale; MetS-AS: Metabolic Syndrome Attitude Scale; MetS-SMBS: Metabolic Syndrome Self-Management Behavior Scale.

**Table 4 ijerph-20-02547-t004:** Correlation between MetS self-management behavior and its variables (n = 231).

	MetS-SMBS	MetS-KS	MetS-AS	Age	TG	No.CD	DBP
MetS-SMBS	1.00						
MetS-KS	0.32 **	1.00					
MetS-AS	0.52 **	0.30 **	1.00				
Age	0.32 **	−0.11	0.21 **	1.00			
TG	−0.16 *	−0.13	−0.20 **	−0.11	1.00		
No.CD	0.25 **	0.01	0.28 *	0.44 **	0.01	1.00	
DBP	−0.20 **	−0.05	−0.15 **	−0.14 *	0.19	0.04	1.00

* *p* < 0.05, ** *p* < 0.01. Note. MetS-SMBS: Metabolic Syndrome Self-Management Behavior Scale; MetS-KS: Metabolic Syndrome Knowledge Scale; MetS-AS: Metabolic Syndrome Attitude Scale; TG: triglyceride level; No.CD: number of chronic diseases; DBP: diastolic blood pressure.

**Table 5 ijerph-20-02547-t005:** Regression analysis of variables predicting MetS self-management behavior total scores (n = 231).

Independent Variable	R	SE	R2	F	p
MetS-AS	0.51	5.88	0.258	85.20	0.000
MetS-KS	0.59	5.54	0.179	14.87	0.000
Age	0.55	5.71	0.158	21.74	0.000
PPC	0.61	5.44	0.032	7.40	0.003
BMI	0.62	5.37	0.023	4.57	0.012

Note. MetS-AS: Metabolic Syndrome Attitude Scale; MetS-KS: Metabolic Syndrome Knowledge Scale; PPC: perceived physical condition; BMI: body mass index.

## Data Availability

The data presented in this study are available on request from the corresponding author.

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
