# Peer review of "Metabolic Syndrome-Related Knowledge, Attitudes, and Behavior among Indigenous Communities in Taiwan: A Cross-Sectional Study"

_ijerph, 2023, doi:10.3390/ijerph20032547_

Round 1
Reviewer 1 Report
The research topic is important, especially in light of the fact that metabolic syndrome (MetS) is a global health issue, and low level of knowledge, attitude and self-management behavior can contribute to increasing the influence MetS risk. The manuscript is well-written and easy to flow.
However, internal consistency is suggested to align the title, aim, results, and conclusions. It may confuse the reader in different sections whether the study was about the factors associated with the knowledge, attitude and self-management behavior (SMBS) for MetS risk, or associations among KAB themselves, or correlations factors affecting specifically MetS-SMBS.
Specific comments are added in the manuscript itself.

Author Response
The internal consistency has been addressed to improve the readability of the study.

Reviewer 2 Report
From the public health perspective, this article shows important information about Metabolic Syndrome culture among aboriginal communities in Taiwan. The methodology is appropriate, the methodology and results have potential broader applicability to other indigenous people. My expectation is that health professionals use this evidence to improve the health conditions of the indigenous community in Taiwan.
Over line 86: to correct text, it says paticiapnts, must say participants.
Over line 171 , Table 1 : may be is better to use Range , also to write on BOLD each group of the sociodemographic characteristics.
Author Response
Point 1: Over line 86: to correct text, it says paticiapnts, must say participants.
Response 1: The word “participants” (Line 86) has been corrected accordingly.
Point 2:Over line 172 , Table 1 : may be is better to use Range , also to write on BOLD each group of the sociodemographic characteristics.
Response 2: The word has been changed to “range,” (Line 179,181)
Reviewer 3 Report
Metabolic syndrome investigation is very important challenge. An important area of study is the influence of MetS on clinical outcomes among people in various weight categories. The paper is very interesting anyway there are a lot of limitations of the study (apart of mentioned ones).
1. The investigation has a local manner and scientific outcome is described very vague. The conclusion of the article which states that the relationship between metabolic syndrome knowledge, attitudes, and behaviors can serve as a reference point for medical administrative agencies in constructing educational content and preventive health treatment is not justified. It seems more recommendation rather than conclusion. Definition of reference point also is not clear and needs to be justified.
2. There is a need to provide the novelty of the investigation.
3. The aim of the study is not described well.
4. Abstract must be changed. It is not providing an opportunity to understand neither the novelty nor the main research outcome.
5. The style of the introduction must be changed. It is not clear at all why the statements are with US. No explanation, no justification. The top 10 causes of death in Taiwan and the United States have changed from infectious to chronic degenerative diseases, particularly cardiovascular diseases (CVD) and diabetes mellitus. What is common in US and Taiwan in order to group them. After US it is making a reference to Iran survey. For me it is really very difficult to understand the selection of the countries and the links with Taiwan. I do dot believe that those countries have the same level of knowledge and information, and, consequently the attitude and behavior.
6. For me data are not up to date. Data were collected from July 01, 2016, to July 31, 2017. In current world where the extremally fast changes of knowledge is available I do not believe that from scientific point of view it is correct to use the data from 2017.
Author Response
Point 1: The investigation has a local manner and scientific outcome is described very vague. The conclusion of the article which states that the relationship between metabolic syndrome knowledge, attitudes, and behaviors can serve as a reference point for medical administrative agencies in constructing educational content and preventive health treatment is not justified. It seems more recommendation rather than conclusion. Definition of reference point also is not clear and needs to be justified.
Response 1:
We have revised the descriptions in the abstract(Line 18-19,37-39), discussion(Line 257), and conclusion(Line 330-332) accordingly.
Point 2: There is a need to provide the novelty of the investigation.
Response 2: The novelty of the investigation is its comparison of those with MetS between genders (Line 201-203; Table 1,2, and 3).
Point 3: The aim of the study is not described well
Response 3: The aim of the study has been revised.
We proposed two study aims: (a) to assess the knowledge of, attitudes, and behavior toward MetS among indigenous Taiwanese people and (b) to determine predictors of MetS self-management behavior among indigenous people(Line 86-89).
Point 4: Abstract must be changed. It is not providing an opportunity to understand neither the novelty nor the main research outcome.
Response 4 : The abstract has been revised accordingly.
Point 5: The style of the introduction must be changed. It is not clear at all why the statements are with US. No explanation, no justification. The top 10 causes of death in Taiwan and the United States have changed from infectious to chronic degenerative diseases, particularly cardiovascular diseases (CVD) and diabetes mellitus. What is common in US and Taiwan in order to group them. After US it is making a reference to Iran survey. For me it is really very difficult to understand the selection of the countries and the links with Taiwan. I do dot believe that those countries have the same level of knowledge and information, and, consequently the attitude and behavior.
Response 5: I have included an article focusing on the Asia-Pacific region in the literature and MetS articles that connect studies from these countries with Taiwan’s economic development and living habits.
Point 6: For me data are not up to date. Data were collected from July 01, 2016, to July 31, 2017. In current world where the extremally fast changes of knowledge is available I do not believe that from scientific point of view it is correct to use the data from 2017.
Response 6: This study was conducted between 2016 and 2017, and the interpretation of the research results depends on time. Few studies have investigated MetS in indigenous communities; our results can serve as reference for future research.
Reviewer 4 Report
General comments
1. English must be corrected by professional language services. In this version it is impossible to accept the manuscript.
2. The data presented in Tables should be presented (and discussed) depending on gender.
Detailed comments
3. Abstract must be significantly shortened (too many details describing the methods and results are presented)
4. Line 44: clear definition of metabolic syndrome should be presented.
5. Line 49: the ATP criteria should be provided.
6. Line 50: not only in Taiwan, please reformulate the sentence.
7. Line 51: please expand these numbers.
8. Line 52: please change from ‘improper high-fat diet’ into ‘imbalanced (e.g. high-fat) diet’.
9. Line 54: please delete the sentence.
10. Lines 55-57: please correct the sentence to clearly match the numbers to the countries.
11. Line 63: please clearly define ‘indigenous people’.
12. Second paragraph: please divide it into shorter sections.
13. Lines 84-85: what is ‘convenience sampling method’?
14. Tables: please change from ‘rang’ to ‘range’.
15. References should be adjusted to the journal’s requirements.
16. Table 2 must be re-organized (it is extremely hard to read the data).
Author Response
Response to Reviewer 4 Comments
Point 1: English must be corrected by professional language services. In this version it is impossible to accept the manuscript.
Response 1: We have sent the manuscript to a professional editing service for revision
Point 2: The data presented in Tables should be presented (and discussed) depending on gender.
Response 2: We have organized the results in the table by gender(Line 201-203ï¼›213-214;Table 1,Table 2,Table3).
Point 3: Abstract must be significantly shortened (too many details describing the methods and results are presented)
Response 3: We have revised the scope of the abstract in accordance with the committee’s suggestions.
Point 4: Line 44: clear definition of metabolic syndrome should be presented.
Response 4: We defined “metabolic syndrome” on lines 124–132 of the original manuscript.
Point 5: Line 49: the ATP criteria should be provided.
Response 5: The ATP criteria are stated in lines 125-132 of the original manuscript.
Point 6: Line 50: not only in Taiwan, please reformulate the sentence.
Response 6: The sentence has been revised; please see lines 54–58.
Point 7: Line 51: please expand these numbers.
Response 7: The numbers have been revised; please see lines 58–59.
Point 8: Line 52: please change from ‘improper high-fat diet’ into ‘imbalanced (e.g. high-fat) diet’.
Response 8: The word has been changed as per the reviewer’s suggestion; please see lines 60-61.
Point 9: Line 54: please delete the sentence.
Response 9: The sentence on Line 62 has been deleted.
Point 10: Lines 55-57: please correct the sentence to clearly match the numbers to the countries.
Response 10: The sentence has been corrected as per the reviewer’s suggestion(Line 64-67).
Point 11: Line 63: please clearly define ‘indigenous people’.
Response 11: This word was misplaced and has been changed accordingly.
Point 12: Second paragraph: please divide it into shorter sections.
Response 12: The paragraph has been divided into shorter sections as per the reviewer’s suggestion.
Point 13: Lines 84-85: what is ‘convenience sampling method’?
Response 13: “Convenience sampling” in this study refers to the research team finding and examining research participants from the indigenous tribe.
Point 14: Tables: please change from ‘rang’ to ‘range’.
Response 14: The spelling of this word has been corrected.
Point 15: References should be adjusted to the journal’s requirements.
Response 15: The references have been adjusted to meet the journal’s requirements.
Point 16: Table 2 must be re-organized (it is extremely hard to read the data).
Response 16: The table has been reorganized accordingly.
Round 2
Reviewer 3 Report
The authors made an effort in order to improve the paper. In this form it can be published.
Author Response
Thanks to the reviewer for your reminders and suggestions.
Reviewer 4 Report
The manuscript requires further corrections before the final acceptation.
1. Multiple typo errors can be found in throughout the manuscript (e.g. please see the references’ formatting). It should be read again with an appropriate editorial care.
2. As indicated before, the abstract is not a detailed description of the study but an abstract. I strongly suggest shortening it by e.g. excluding methodological information which should not be presented in this part of the manuscript.
3. The metabolic syndrome should be defined at the beginning of the manuscript (=introduction). The same as ATP criteria.
4. Please use the same font in all tables.
5. Table 2 should be also re-organized to make it easy to be followed.
Author Response
Point 1: Multiple typo errors can be found in throughout the manuscript (e.g. please see the references’ formatting). It should be read again with an appropriate editorial care.
Response 1: The manuscript has been revised accordingly.
Point 2:As indicated before, the abstract is not a detailed description of the study but an abstract. I strongly suggest shortening it by e.g. excluding methodological information which should not be presented in this part of the manuscript.
Response 2: The abstract has been revised accordingly.
Point 3:The metabolic syndrome should be defined at the beginning of the manuscript (=introduction). The same as ATP criteria.
Response 3: We have revised the metabolic syndrome definition in the introduction section(Line 86).
Point 4: Please use the same font in all tables.
Response 4: We have revised the same font in all tables.
Point 5:Table 2 should be also re-organized to make it easy to be followed.
Response 5: The table 2 has been reorganized accordingly.